# Transport Numbers and Electroosmosis in Cation-Exchange Membranes with Aqueous Electrolyte Solutions of HCl, LiCl, NaCl, KCl, MgCl_2_, CaCl_2_ and NH_4_Cl

**DOI:** 10.3390/e27010075

**Published:** 2025-01-15

**Authors:** Simon B. B. Solberg, Zelalem B. Deress, Marte H. Hvamstad, Odne S. Burheim

**Affiliations:** Department of Energy and Process Engineering, Norwegian University of Science and Technology, NO-7491 Trondheim, Norway; zelalem.b.deress@ntnu.no (Z.B.D.); odne.s.burheim@ntnu.no (O.S.B.)

**Keywords:** ion-exchange membrane, electrodialysis, permselectivity, electroosmosis, transport number

## Abstract

Electroosmosis reduces the available energy from ion transport arising due to concentration gradients across ion-exchange membranes. This work builds on previous efforts to describe the electroosmosis, the permselectivity and the apparent transport number of a membrane, and we show new measurements of concentration cells with the Selemion CMVN cation-exchange membrane and single-salt solutions of HCl, LiCl, NaCl, MgCl_2_, CaCl_2_ and NH_4_Cl. Ionic transport numbers and electroosmotic water transport relative to the membrane are efficiently obtained from a relatively new permselectivity analysis method. We find that the membrane can be described as perfectly selective towards the migration of the cation, and that ${\mathrm{Cl}}^{-}$ does not contribute to the net electric current. For the investigated salts, we obtained water transference coefficients, $t_{w}$, of 1.1 ± 0.8 for HCl, 9.2 ± 0.8 for LiCl, 4.9 ± 0.2 for NaCl, 3.7 ± 0.4 for KCl, 8.5 ± 0.5 for MgCl_2_, 6.2 ± 0.6 for CaCl_2_ and 3.8 ± 0.5 for NH_4_Cl. However, as the test compartment concentrations of LiCl, MgCl_2_ and CaCl_2_ increased past 3.5, 1.3 and 1.4 mol kg^−1^, respectively, the water transference coefficients appeared to decrease. The presented methods are generally useful for characterising concentration polarisation phenomena in electrochemistry, and may aid in the design of more efficient electrochemical cells.

## 1. Introduction

Ion-exchange membranes are essential components of reverse electrodialysis power plants; they provide the means to mix fresh water and seawater in a controlled manner such that electric power can be extracted from the cell. The opposite process, electrodialysis, is also showing promise for use in wastewater treatment [1,2,3], water desalination [4], and demineralisation in food processing [5,6,7,8,9,10,11,12,13]. Cheap, selective membranes with low resistance are necessary for these processes [14,15]. As such, it is crucial to have accurate experimental methods and meaningful performance indicators for describing the transport phenomena in ion-exchange membranes.

Permselectivity is frequently used to describe a membrane’s selectivity towards an ionic species, but several different definitions are used in the literature [16]. It has been shown that a particularly useful definition is the ratio of the measured electric potential across an ion-exchange membrane divided by the chemical potential difference of the salt in question, since it directly relates the permselectivity to the ionic transport numbers and the neutral component transference coefficients [17,18,19,20]. In aqueous electrolyte mixtures, the water transference coefficient is of particular interest, since it describes the electroosmotic transport of water dragged along as a hydration shell around charge-transporting ions [21,22]. The electroosmotic transport affects the amount of work involved in the movement of ions through the ion-exchange membrane, usually lowering the permselectivity and the reversible voltage difference across the membrane [19,23]. Both the transport numbers and the transference coefficients of the membrane are useful concepts for the description of ion-exchange membrane selectivity.

Water transference coefficients, $t_{w}$, and electroosmosis have been investigated for perfluorinated Nafion membranes in single-salt electrolyte mixtures, mainly by use of streaming potential measurements [24,25,26,27,28,29]. Examples of measured water transference coefficients for the Nafion 117 membrane are *t_w_* ≈ 2.6, ≈5 and ≈15 for chloride salts of H^+^, K^+^ and ${\mathrm{Mg}}^{2+}$, respectively. This corresponds to 2.6, 5 and 30 moles of water per mole of cation transporting charge [21]. Clearly, the magnitude of electroosmotic transport can be large. Hydrophobic and hydrophilic polymers will likely facilitate different water transference coefficients, and these factors may be important for the design of efficient membranes for a particular process. Ionic transport numbers and water transference coefficients have also been extracted from electric potential differences across membranes through various methods of analysis [17,30]. We shall here further contribute to the discussion around strategies for analysing membrane potentials, with a focus on transport numbers and transference coefficients, by using a relatively novel permselectivity method.

We present new measurements of membrane potentials of the commercial cation-exchange membrane Selemion CMVN. These new electric potential measurements are for membranes in contact with aqueous single-salt solutions of HCl, LiCl, NaCl, MgCl_2_, CaCl_2_ and NH_4_Cl. Furthermore, we shall use the novel permselectivity analysis method communicated by Solberg et al. [19] in order to determine ionic transport numbers and water transference coefficients for the Selemion CMVN equilibrated with these salts. The analysis method is derived from the theory of non-equilibrium thermodynamics, in which the fluxes and driving forces are obtained from the entropy production.

## 2. Thermodynamic Framework

### 2.1. The Measurable Electric Potential

The electric potential difference across the cation-exchange membrane was measured in a concentration cell, as illustrated in Figure 1. We used two Ag/AgCl electrodes, and the two electrolyte solution compartments with different salt concentrations were thoroughly stirred. The main contributions to the total cell voltage were therefore due to the salt concentration gradient across the membrane, as well as a small contribution from the differences in chemical potentials of the solid phases of the electrodes (the electrode bias) [21]. Measurements were taken under isobaric and isothermal conditions, and we aimed to achieve a pseudo-steady state where the membrane interfaces were in a Donnan equilibrium. Transport phenomena in the ion-exchange membrane of such a cell for a laboratory frame of reference were thoroughly described by Solberg et al. using non-equilibrium thermodynamics [20]. We repeat only the essential parts here.

Under open-circuit conditions, i.e., negligible electric current density, the measurable electric potential difference, Δϕ, across the ion-exchange membrane is [20,21]: (1)$$\begin{array}{c} -F\Delta \phi =\frac{t_{M^{z+}}}{z^{+}}\Delta \mu_{MCl_{z+}}+t_{w}\Delta \mu_{w} \end{array}$$
where *F* is the Faraday constant, $t_{M^{z+}}$ and $z^{+}$ are the transport number and charge of the cation and $t_{w}$ is the water transference coefficient. Furthermore, $\Delta \mu_{w}$ is the chemical potential difference of water, and $\Delta \mu_{MCl_{z+}}$ is the chemical potential difference of the neutral salt of the $M^{z+}$ cation consisting of $z^{+}$ chloride ions. The transport number and the water transference coefficient take on a mean value on the intervals $\Delta \mu_{MCl_{z+}}$ and $\Delta \mu_{w}$, respectively. They are defined as: (2)$$t_{M^{z+}}=z^{+}F{\left(\frac{J_{M^{z+}}}{j}\right)}_{d\mu =0},\;\;t_{w}=F{\left(\frac{J_{w}}{j}\right)}_{d\mu =0}$$
where $j=F\left(z^{+}J_{M^{z+}}-J_{{Cl}^{-}}\right)$ is the current density and $J_{M^{z+}}$, $J_{{Cl}^{-}}$ and $J_{w}$ are the molar fluxes of cations, chloride and water, respectively. The subscript dμ=0 refers to a uniform composition. It is essential for the definition of the transport numbers, but these quantities retain their physical meaning even outside of uniform composition. We make a distinction between the ionic transport number and the water transference coefficient according to the naming convention used by Kjelstrup and Bedeaux [21]. Transport numbers are here reserved for ions contributing to the electric current, while transference coefficients describe the amount of a neutral component that is indirectly transported by the current. Electroosmosis is the relevant example, describing the amount of water coupled to the movement of migrating ions.

We evaluate the two chemical potentials in terms of one state variable, i.e., the molality. It is related to the number of moles of salt, $n_{MCl_{z+}}$, and the moles of water, $n_{w}$, through: (3)$$m_{MCl_{z+}}=\frac{n_{MCl_{z+}}}{M_{w}n_{w}}$$
where $M_{w}$ is the molecular weight of water. With this state variable choice, in which the molality is the only independent variable, we define the chemical potentials as [31]: (4)$$\begin{array}{cc} \mu_{MCl_{z+}} & =\mu_{MCl_{z+}}^{0}+RT\ln\left(\frac{m_{M^{z+}}m_{{Cl}^{-}}^{\nu +}\gamma_{MCl_{z+}}^{\nu }}{m_{M^{z+}}^{0}{\left(m_{{Cl}^{-}}^{0}\right)}^{\nu +}}\right)\\ \mu_{w} & =\mu_{w}^{0}+RT\ln a_{w} \end{array}$$
where *R* is the gas constant, *T* is the absolute temperature, $\gamma_{MCl_{z+}}$ is the mean molal activity coefficient, $a_{w}$ is the water activity and the factors $\nu =\nu^{+}+\nu^{-}$ represent the total number of ionic species, and the individual cationic and anionic species produced by the dissolution of a salt. Here, $\nu^{-}=1$ since all investigated components are chloride salts. The superscript 0 denotes the reference state of the chemical potentials, here referring to infinitely dilute salt in water.

We shall now use a permselectivity definition consisting of the measured electric potential divided by the chemical potential difference of the salt. This is a common way to quantify a membrane “permselectivity” metric [18,19,32] that has the added benefit of scaling the measured electric potential such that large electric potentials do not introduce bias in the regression analysis. The resulting permselectivity can be understood in terms of either the apparent transport number or via the permselectivity analysis shown by Solberg et al. [19].

### 2.2. The Apparent Transport Number

The chemical potential differences of water and salt in the two-component mixture are related through the Gibbs–Duhem equation: $d\mu_{w}=-m_{M^{z+}}M_{w}d\mu_{MCl_{z+}}$ [21]. Here, $m_{MCl_{z+}}=m_{M^{z+}}$ denotes the cation molal concentration. This relation is frequently used to reduce the number of driving forces that need to be calculated [17], such that the measurable electric potential becomes: (5)$$\begin{array}{c} -F\Delta \phi =\left(\frac{t_{M^{z+}}}{z^{+}}-{\bar{m}}_{M^{z+}}M_{w}t_{w}\right)\Delta \mu_{MCl_{z+}} \end{array}$$The overhead bar, ${\bar{m}}_{M^{z+}}$, signifies that the molal concentration takes on a mean value as the expression is integrated over the $\Delta \mu_{MCl_{z+}}$ interval. This mean value need not be the arithmetic mean of the reference and test concentrations used. The term wrapped in parentheses is frequently referred to as the apparent transport number, $t_{M^{z+}}^{\mathrm{app}}$, and it is found by evaluating the permselectivity according to: (6)$$\begin{array}{c} t_{M^{z+}}^{\mathrm{app}}=-F\left(\frac{\Delta \phi }{\Delta \mu_{MCl_{z+}}}\right)=\left(\frac{t_{M^{z+}}}{z^{+}}-{\bar{m}}_{M^{z+}}M_{w}t_{w}\right) \end{array}$$This equation is sometimes also referred to as the Scatchard equation [33,34]. The method states that a decrease in the membrane permselectivity is due to a reduction in the apparent transport number, i.e., the ionic transport number corrected for water transport.

### 2.3. The Permselectivity

The permselectivity may also be evaluated without invoking the Gibbs–Duhem equation [19]. We have: (7)$$\alpha =-F\left(\frac{\Delta \phi }{\Delta \mu_{MCl_{z+}}}\right)=\frac{t_{M^{z+}}}{z^{+}}+t_{w}\left(\frac{\Delta \mu_{w}}{\Delta \mu_{MCl_{z+}}}\right)$$
where α is the permselectivity. The permselectivity and the apparent transport number are, in fact, the same quantity, but we use separate symbols to denote the two different analysis methods. For the permselectivity analysis, the transport number, $t_{MCl_{z+}}$, and the transference coefficient, $t_{w}$, are used as constant regression coefficients.

## 3. Data Reduction

The activity coefficient data necessary for the calculation of the chemical potentials of salt and water were gathered from the literature [31,35] and fitted to the Pitzer equations [36]. See the Appendix A for a summary of the values used in the analysis.

### 3.1. The Apparent Transport Number

The apparent transport number of the cation-exchange membrane was found by first fitting an empirical function to the measured electric potential. The reported electric potentials are monotonically increasing functions of the chemical potential difference of the salt. Non-linear S-shaped functions such as the hyperbolic tangent or the error function are therefore useful. Other choices, such as polynomial functions, could introduce extrema in the apparent transport number function. We used the following function: (8)$$-F\Delta \phi =\frac{A\Delta \mu_{MCl_{z+}}}{\sqrt{1+B{\left(\Delta \mu_{MCl_{z+}}\right)}^{2}}}+\epsilon_{\phi }$$
where *A* and *B* are regression coefficients, and $\epsilon_{\phi }$ is the regression residual error. This provides an empirical relation for the apparent transport number: (9)$$t_{M^{z+}}^{\mathrm{app}}=-F\left(\frac{\Delta \phi }{\Delta \mu_{MCl_{z+}}}\right)=\frac{A}{\sqrt{1+B{\left(\Delta \mu_{MCl_{z+}}\right)}^{2}}}$$

### 3.2. The Permselectivity

The permselectivity was evaluated in terms of a linear function based on Equation (Equation 7) using the ionic transport number and the water transference coefficient as the regression coefficients.(10)$$\alpha =-F\left(\frac{\Delta \phi }{\Delta \mu_{MCl_{z+}}}\right)=C+D\left(\frac{\Delta \mu_{w}}{\Delta \mu_{MCl_{z+}}}\right)+\epsilon_{\alpha }$$
where $\epsilon_{\alpha }$ is the regression residual, while $C=t_{M^{z+}}/z^{+}$ and $D=t_{w}$ are the regression coefficients. In the case of the ionic transport numbers or the water transference coefficients varying with the salt concentration, the regression model will be found lacking. This was the case for LiCl, MgCl_2_ and CaCl_2_, when the concentration of the salt in the test compartment increased past 3.5, 1.3 and 1.4 mol ${\mathrm{kg}}^{-1}$, respectively. Therefore, only measurements with concentrations below these limits, corresponding to a driving force of $\Delta \mu_{MCl_{z+}}=20$ kJ ${\mathrm{mol}}^{-1}$, were used in the permselectivity analysis.

The resulting ionic transport numbers and water transference coefficients were then used together with Equation (Equation 9) to estimate the mean molal concentration across the two electrolyte solutions according to the following: (11)$${\bar{m}}_{M^{z+}}=\frac{\frac{t_{M^{z+}}}{z^{+}}-t_{M^{z+}}^{\mathrm{app}}}{M_{w}t_{w}}$$The resulting mean concentration represents the contribution from the chemical potential difference of water, and it may deviate from the arithmetic mean of the test and reference solution concentrations.

## 4. Experimental Methods

Aqueous electrolyte solutions of chloride salts were prepared gravimetrically using demineralised water with an electrical conductivity of $5.5\;\mu S\;m^{-1}$. Of the alkali metal salts, LiCl (99%, Thermo Fisher Scientific, Waltham, MA, USA) and NaCl (≥99%, Sigma-Aldrich, St. Louis, MO, USA) were investigated. The KCl results from the same concentration cell and cation-exchange membrane were obtained by Solberg [20], and they were included in this analysis. Of the alkaline earth metals, MgCl_2_ · 6H_2_O (AnalaR NORMAPUR, VWR Chemicals, Leicestershire, UK) and CaCl_2_ · 2H_2_O (AnalaR NORMAPUR, VWR Chemicals) were investigated. In addition, HCl (32%*w*/*w* EMSURE, Merck, Darmstadt, Germany) and NH_4_Cl (ACS, Merck) were used. The molal concentrations used for the test solutions are shown in Table 1.

The Selemion CMVN cation-exchange membrane (AGC Engineering Co., Ltd., Chiba, Japan) was delivered in the Na^+^ form. Appropriately sized pieces of the membrane were stored in corresponding electrolyte solutions of 0.1 mol ${\mathrm{kg}}^{-1}$ for at least one week before use. The electrolyte solution was refreshed once during this week to ensure the absorption and equilibration of the target cation. A thickness of roughly 100 μm, a water uptake of 0.182 g $g_{\mathrm{polymer}}^{-1}$ and an ion-exchange capacity of 1.40 meq $g_{\mathrm{polymer}}^{-1}$ are typical for these commercial membranes [37].

Ag/AgCl reference electrodes were fabricated from two polished silver rods of suitable length (≥99.95% Ag, 5 mm diameter, Thermo Fisher Scientific). The silver rods were electroplated in a 0.1 mol ${\mathrm{kg}}^{-1}$ diluted solution of HCl (32%*w*/*w* EMSURE, Merck) in order to form a silver chloride coating. An electric current of 1 mA was passed using Pt as the counter electrode over the course of 12 h. The Ag/AgCl electrodes were stored short-circuited in the aqueous 0.1 mol ${\mathrm{kg}}^{-1}$ HCl solution when not in use in order to minimise the bias potential between them. A Gamry Interface 5000E potentiostat was used for all logging of electric potentials and electroplating (with a declared standard deviation, σ, for the measured voltage in mV of 2σ=1+0.003×Δϕ). The bias potential was found as the lower limit of the open-circuit potential measured over a period of 300–600 s between the two reference electrodes exceeded 1 mV when situated in the same 0.1 mol ${\mathrm{kg}}^{-1}$ electrolyte solution. It was measured directly after measuring the electric potential of the concentration cell, and some (concentration-dependent) capacitance was evident as an initial quick drop in voltage and stabilisation around the true electrode bias potential. The silver chloride coating was refreshed if the measured electrode bias potential exceeded 1 mV.

The concentration cell used for the measurement of electric potentials across the cation-exchange membrane, illustrated in Figure 1, consisted of two compartments of roughly 300 mL. Each compartment was filled with 200 mL of electrolyte solution. The left-hand side of Figure 1 is denoted as the reference solution, while the right-hand side is referred to as the test solution. A cation-exchange membrane with an exposed area of $7.10\;{\mathrm{cm}}^{2}$ separated the two compartments, preventing mixing of the two electrolyte solutions. The left-hand side of the cell always held a concentration of 0.1 mol ${\mathrm{kg}}^{-1}$ of the salt to be tested. The electrolyte solution concentrations on the right-hand side were varied. Both compartments were thoroughly stirred with magnetic stirrers to minimise the contribution from concentration polarisation in the boundary layers adjacent to the membrane. With one Ag/AgCl reference electrode in each compartment, the open-circuit potential of the concentration cell was measured over the course of at least 1800 s. Each concentration level was measured two times with fresh electrolyte solutions. These membrane potentials were measured under isobaric conditions (maintained by ensuring equal heights for the two solutions) at an ambient temperature of 295 ± 1 K, and the final value was taken as the mean of a stable reading of at least 900 s. A negative drift was generally not present in the measured potential, indicating that diffusive transport of both water (osmosis) and/or salt did not significantly alter the electrolyte solution concentrations over the course of the measurement. HCl, however, did show, in initial tests, a clearly visible drift that was more pronounced at high concentration gradients across the membrane. Therefore, 10 cation-exchange membranes (with a total thickness of roughly 1 mm) were used for the reported results HCl, while 1 membrane was sufficient for the other salts.

## 5. Results and Discussion

The measured electric potentials of the concentration cell, as illustrated in Figure 1, were dominated by the contribution from the concentration gradient across the cation-exchange membrane. In the observed electric potential, a small contribution was present due to differences in the chemical potentials of the solid phases of the electrodes (the electrode bias). This contribution was measured and subtracted from the reported electric potentials, but it was generally below 1 mV. The measured potentials are summarised together with thermodynamic data for the salts in the Appendix A. Concentration gradients in the bulk electrolyte solutions were minimised by rigorous stirring, such that its contribution to the observed potential was negligible. The variance of each individual measurement was low, such that the double standard deviations fell within the scatter plot boxes and error bars were therefore left out of subsequent figures.

The methods applied in this work can be leveraged to measure particularly useful performance indicators for ion-exchange membranes; i.e., the transport numbers and electroosmosis. Such results may later be combined with measurements of the electric conductivity and diffusion coefficients to fully characterize the membrane transport phenomena in terms of the phenomenological coefficients [21]. This may then be compared with equilibrium properties for different membranes, such as water uptake, polymer type, ion-exchange capacity and the nature of the membrane fixed ion groups, in order to better understand these materials. Here, however, we focus on the determination of transport numbers and transference coefficients.

### 5.1. The Apparent Transport Number

The electric potential across the Selemion CMVN cation-exchange membrane is shown for HCl, LiCl, NaCl and MgCl_2_ in Figure 2a. The other salts have been left out of this figure to enhance legibility, but equivalent trends are observed. The non-linear regression provides a good description of the electric potential as a function of the chemical potential, but there are still data trends that are unexplained by these non-linear models. This is illustrated in Figure 2b, where the regression residuals of NaCl and MgCl_2_ are shown. The variance in repeated measurements is small, but residuals vary much more as the experimental conditions are varied. Errors in the calculation of the chemical potentials, where both activity coefficients and the molal concentration carry some uncertainty, could contribute to the model’s inability to explain all trends. Nevertheless, the regression residuals are below 2.5% of the electric potential in all cases, such that the model provides a good estimate of the apparent transport numbers of the salts.

Individual salt apparent transport numbers, found from the non-linear regression models divided by the chemical potential difference of the salts, are shown in Figure 3. For salts of monovalent cations, they trend toward $t_{MCl_{z+}}^{\mathrm{app}}=1$ as the chemical potential difference across the membrane moves toward zero. They trend toward $t_{MCl_{z+}}^{\mathrm{app}}=0.5$ for divalent cation salts, since the cation valency is included in this apparent transport number definition. This indicates perfect cation selectivity for the membrane, $t_{M^{z+}}=1$ and $t_{{Cl}^{-}}=0$, in the case where the whole concentration cell has uniform composition. The subsequent decrease in the apparent transport number as the salt’s driving force increases can be understood as either a reduction in the cation selectivity, an increased influence of water transference, or a combination of both.

The physical significance of the water transference is understood as follows. Consider a spatial direction in which a concentration gradient of one aqueous salt is positive across the membrane, leading to a positive chemical potential gradient and a positive value of −FΔϕ. This means that the concentration cell can perform useful work in the external circuit, as long as the concentration difference persists. The chemical potential gradient of the water, however, is negative in this direction. Any water molecules transferred along with migrating cations—that is, electroosmosis—are transported in a direction that increases their chemical potential. This must decrease the available energy from the transport of cations from high to low concentration. By using Equation (Equation 1), this effect is fully captured by $t_{w}\Delta \mu_{w}$. By applying the Gibbs–Duhem equation, resulting in Equation (Equation 5), the water transference effect is instead captured by the apparent transport number, $t_{MCl_{z+}}^{\mathrm{app}}$. When electric current is applied to the cell, desalinating one compartment by driving ions from low to high concentration, the effect is reversed.

It is important to note that the ionic transport numbers, $t_{M^{z+}}+t_{{Cl}^{-}}=1$, are independent of any diffusive transport that is not accompanied by a net electric current in the external circuit. This means that it is possible for Cl^−^ to leak through the membrane in the same direction as $M^{z+}$ even if $t_{M^{z+}}=1$, but such leakage must be accompanied by an equal amount of $M^{z+}/z^{+}$ such that there is no net charge transport.

The exact magnitude of the water transference coefficient is difficult to extract from the apparent transport number, since it requires knowledge of the mean cation concentration across the $\Delta \mu_{MCl_{z+}}$ interval. This value may be different from the arithmetic mean of the two external solution concentrations. In the continuation of the analysis, it is therefore useful to apply the permselectivity model from Equation (Equation 7). This may allow for the direct identification of membrane water transference coefficients.

### 5.2. The Membrane Permselectivity

The permselectivity values of all of the investigated salts are shown in Figure 4. All monovalent cation salts, except for LiCl, show a linear permselectivity decrease as the ratio of chemical potentials is increased, and a lack-of-fit statistic [19,38] shows that statistically significant trends have been captured. It is possible that this decrease is a result of a decrease in cation selectivity; i.e., $t_{M^{z+}}\neq 1$. However, a simpler explanation is that the permselectivity follows Equation (Equation 7) with constant transport numbers, where $t_{w}$ is the slope of the linear regression and $t_{M^{z+}}/z^{+}$ is the *y*-axis intercept. The obtained regression values are presented in Table 2. This model can fully explain the variation in the membrane permselectivity for most salts, but deviations from linearity arise for hygroscopic salts.

For the hygroscopic salts LiCl, MgCl_2_ and CaCl_2_, there is a systematic deviation from the linear behaviour when the chemical potential difference of the salt exceeds $\Delta \mu_{MCl_{z+}}=20$ kJ ${\mathrm{mol}}^{-1}$. With a reference compartment molality of 0.1 mol ${\mathrm{kg}}^{-1}$, this corresponds to the test molalities of around 3.5, 1.3 and 1.4 mol ${\mathrm{kg}}^{-1}$ for LiCl, MgCl_2_ and CaCl_2_, respectively. Nevertheless, we used the linear region below this in order to estimate the ionic transport numbers and water transference coefficients, as shown in Table 2. The ionic transport numbers are close to and have double standard deviation intervals that include unity, indicating that the Selemion CMVN cation-exchange membrane is adequately described by perfect cation selectivity. Any reduction in the permselectivity in the linear region can instead be explained by the magnitude of electroosmosis.

The non-linear region of the LiCl, MgCl_2_ and CaCl_2_ curves in Figure 4 is more complicated. The slopes become shallower, which could in part be due to $t_{M^{z+}}\neq 1$. In that case, Cl^−^ will carry charge and water in the opposite direction of M^z+^, which leads to a reduction in the net water transference coefficient that is observed. On the other hand, the behaviour could also be explained exclusively by a reduction in the water transference coefficient while $t_{M^{z+}}=1$. It may be necessary to measure the salt concentration over time during electrodialysis operation to fully characterise the transport coefficients in this non-linear regime. Measuring the permselectivity with both the test and reference compartment in the highly concentrated regime for these hygroscopic salts is a natural choice for the continuation of this work. Such measurements could verify if the ionic transport number decreases at high mean concentrations, and contribute to the construction of models for the concentration dependencies of the transport numbers and transference coefficients.

The water transference coefficient of H^+^, $t_{w}=1.1\pm 0.8$, suggests that migrating protons generally enter and leave the membrane as H_3_O^+^ ions. Furthermore, the low magnitude is similar to the results of Okada et al. for the Nafion 117 [25], and suggests that proton migration in the Selemion CMVN occurs in part via the Grotthuss mechanism as reported for the Nafion 117.

With the exception of HCl, there is a definite trend of decreasing water transference coefficients, $t_{w}$, as one moves down a periodic table group. The same trend was observed for Li^+^, Na^+^ and K^+^ in Nafion 117 membranes by Okada et al. and Xie et al. [29,39], as well as for the polymeric ion-exchange membranes Neosepta CM2 by Larchet et al. [34], and for MC-40, MC-41, MA-41 and similar types by Berezina et al. [40]. Berezina and coworkers measured the volume flow as electric current was passed through the membrane, but they seemingly did not correct for the effect of the partial molar volume of the dissolved salt [40]. The reported decreasing water transference coefficient as the external solution salt concentration increases may therefore be overestimated, especially at high concentrations of salt. Without correcting for the partial molar volume of the salt, the value obtained is instead a volumetric permeability, as calculated by Barragán et al. for the Nafion 117 [41]. Larchet and coworkers did correct for the partial molar volume of the salt, but scaled its contribution by the ionic transport number instead of the apparent transport number. This choice may explain the concentration dependency observed by Larchet et al. [34]. We did not observe such a concentration dependency for $t_{w}$ of the Selemion CMVN with Na^+^ and K^+^, but a dependency appears to be present for Li^+^, cf. Figure 4. Future studies should investigate the consistency of the water transference coefficient obtained from volume flow, streaming potential and electromotive force methods in more detail.

The water transference trend coincides with the measurements of the cation radii of hydration for bulk electrolyte solutions communicated by Nightingale [42]. The relationship is shown in Figure 5, where the results are also compared with the water transference coefficients obtained by Okada et al. and Xie et al. for the Nafion 117 membrane [29,39]. It is striking how the trends observed for the water transference coefficient of Selemion CMVN are also seen in the Nafion 117, but shifted to higher values. The water transference coefficient trend lines are seemingly discontinuous when going from monovalent to divalent cations, but this is mainly due to the water transference coefficient being defined per mole of electrons transferred in the external circuit. Evaluating $z^{+}t_{w}$ yields a trend that monotonically increases as $r_{h}$ increases, as shown in Figure 5b. Obtaining ionic transport numbers and water transference coefficients of many more types of ion-exchange membranes may aid in the development of general models for transport phenomena.

The water transference coefficients were correlated with the membrane water uptake, which, for the Nafion 117 equilibrated with the relevant salts, was in the range of 10–25 mol ${\mathrm{mol}}_{SO3^{-}}^{-1}$ [26]. The Selemion CMVN has a water uptake of around 0.182 g $g_{\mathrm{polymer}}^{-1}$ and an ion-exchange capacity of 1.40 mmol $g_{\mathrm{polymer}}^{-1}$ [37], leading to 7 mol ${\mathrm{mol}}_{SO3^{-}}^{-1}$. Nafion 117 membranes typically experience swelling in aqueous solutions, while the Selemion CMVN polymer is cross-linked and relatively dense.

Both the water transference coefficient and the ionic mobility of species in the perfluorinated Nafion 117 membrane were investigated by Okada et al. [26]. The magnitude of the water transference coefficient was correlated with the ionic mobility of a species in Nafion 117 membranes, where cations such as Li^+^ showed lower mobility than Na^+^ [26]. In the interior of the membrane, larger hydration shells can contribute to the friction a cation experiences during migration. The water transference coefficient, as well as the membrane hydrophobicity, can therefore be important parameters to optimise for the efficiency and selectivity of ion-exchange membranes.

The definition of the apparent transport number, Equation (Equation 6), suggests that the water transference coefficient can be found from the apparent transport number if the ionic transport number and the average cation molality are known. The permselectivity analysis further suggests that $t_{M^{z+}}=1$, but the cation molality can be difficult to estimate because it takes on a mean value in the $\Delta \mu_{MCl_{z+}}$ interval. It is therefore of interest to estimate this mean cation molality and compare it to the arithmetic mean of the external concentrations.

The average cation concentration across the membrane was estimated using the apparent transport numbers together with the ionic transport number and water transference coefficient from the linear region of the permselectivity. Concentrations for some salts are shown in Figure 6, but the same trend was also observed for the remaining salts. For comparison, the Selemion CMVN features a water uptake of around 0.182 g $g_{\mathrm{polymer}}^{-1}$, and an ion-exchange capacity of 1.40 mmol $g_{\mathrm{polymer}}^{-1}$ [37]. This leads to a molal concentration of charged functional groups of around 7.7 mol ${\mathrm{kg}}^{-1}$. It is evident that the mean concentration in question is not related to the internal concentration in the membrane, but rather a mean of the external states. We observe that the mean value of the cation molality across the membrane phase is different from and smaller than the arithmetic mean of the external concentrations. If the arithmetic mean of the two external concentrations is used for the determination of the water transference coefficient from the apparent transport number, the resulting values may be systematically smaller than the true water transference coefficient of the membrane.

## 6. Conclusions

This work builds on previous efforts to characterise ion selectivity and electroosmosis for ion-exchange membranes, in which non-equilibrium thermodynamics is used to describe the observed electric potential across a membrane in a concentration cell. New measurements for a cation-exchange membrane in contact with single-salt solutions of HCl, LiCl, NaCl, MgCl_2_, CaCl_2_ and NH_4_Cl are presented and analysed.

Evaluating the permselectivity as a function of the ratio of the chemical potentials of water and salt, we find that the Selemion CMVN membrane appears to be perfectly selective to the cation in single-salt solutions, i.e., the transport of Cl^−^ does not contribute to the electric potential and the net current in the external circuit. Instead, any reduction in permselectivity can be explained by the influence of water transference. For the alkali metals LiCl, NaCl and KCl, we obtain water transference coefficients of $t_{w}=9.2$ (±0.8), 4.9 (±0.2) and 3.7 (±0.4), respectively. For the two investigated alkaline earth metals MgCl_2_ and CaCl_2_, coefficients of $t_{w}=8.5$ (±0.5) and 6.2 (±0.6) are observed. For HCl and NH_4_Cl, the values are $t_{w}=1.1$ (±0.8) and 3.8 (±0.5). For LiCl, MgCl_2_ and CaCl_2_, the water transference coefficients appear to decrease as the test compartment concentrations are increased past 3.5, 1.3 and 1.4 mol ${\mathrm{kg}}^{-1}$, respectively. This work suggests that the water transference coefficients remain constant for most salts, but future studies should investigate the concentration dependence of the water transference coefficients of the hygroscopic cations at high concentrations. The magnitude of the water transference coefficients affects the amount of work involved in the transport of ions across the membrane, and it is therefore an important parameter to consider for the design of highly selective and conductive ion-exchange membranes.

## Figures and Tables

**Figure 1 entropy-27-00075-f001:**
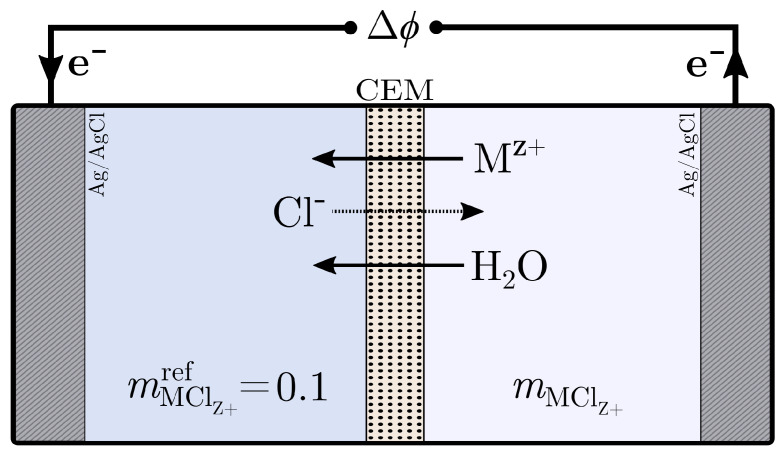
Sketch of the concentration cell used for the cation-exchange membrane (CEM) electric potential measurements, Δϕ. The composition is described by the molality, $m_{MCl_{z+}}$, and the superscript “ref” denotes the reference compartment which always has a salt concentration of $m_{MCl_{z+}}^{\mathrm{ref}}=0.1$. Arrows show the ionic species transport directions that contribute to the electron direction displayed and the measured net electric potential.

**Figure 2 entropy-27-00075-f002:**
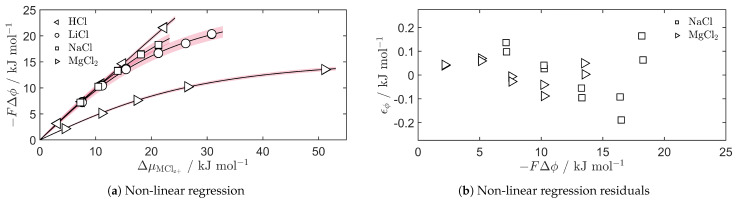
(**a**) Non-linear regression (solid lines) of the measured electric potential, Δϕ, as a function of the chemical potential difference across the membrane, $\Delta \mu_{MCl_{z+}}$, of the salts HCl (left-facing triangles), LiCl (circles), NaCl (squares) and MgCl_2_ (right-facing triangles). Red shaded areas show the 95% confidence interval of the regression curves. (**b**) The regression residuals, $\epsilon_{\phi }$, of the non-linear curves of NaCl and MgCl_2_ illustrating the difference in variance of the repeated measurements compared to between different measurements.

**Figure 3 entropy-27-00075-f003:**
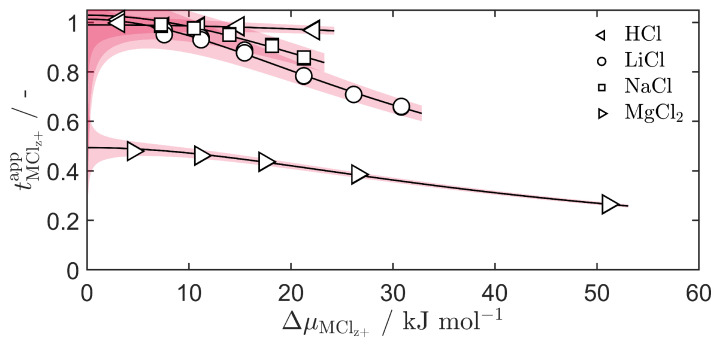
The apparent transport numbers, $t_{MCl_{z+}}^{\mathrm{app}}$, as a function of the chemical potential difference across the membrane, $\Delta \mu_{MCl_{z+}}$, of the salts HCl (left-facing triangles), LiCl (circles), NaCl (squares) and MgCl_2_ (right-facing triangles). Red shaded areas show the 95% confidence interval of the regression curves.

**Figure 4 entropy-27-00075-f004:**
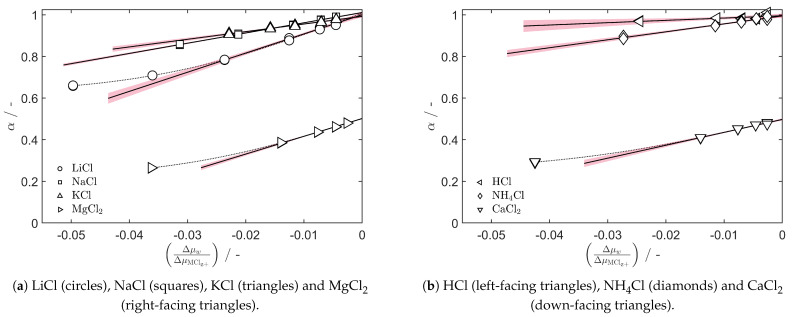
Linear regression (solid lines) of the permselectivity, α, as a function of the chemical potential ratio, $\Delta \mu_{w}/\Delta \mu_{MCl_{z+}}$ in the linear (low-salt-concentration) region. Red shaded areas show the 95% confidence interval of the linear regressions, and non-linear curves (dotted lines) show the deviation from linearity at high salt concentrations for some salts.

**Figure 5 entropy-27-00075-f005:**
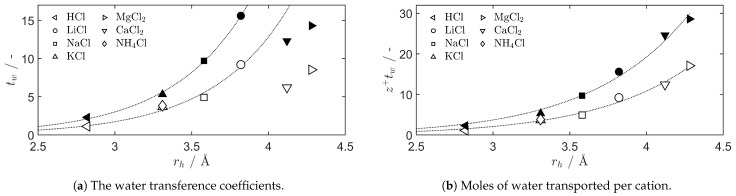
The membrane water transference coefficients, $t_{w}$, from the linear region of the curves of Figure 4, compared to the hydrated cation radius in bulk solutions, $r_{h}$, communicated by Nightingale [42]. The filled symbols show the values for Nafion 117 [29,39], and the dotted lines show general trends.

**Figure 6 entropy-27-00075-f006:**
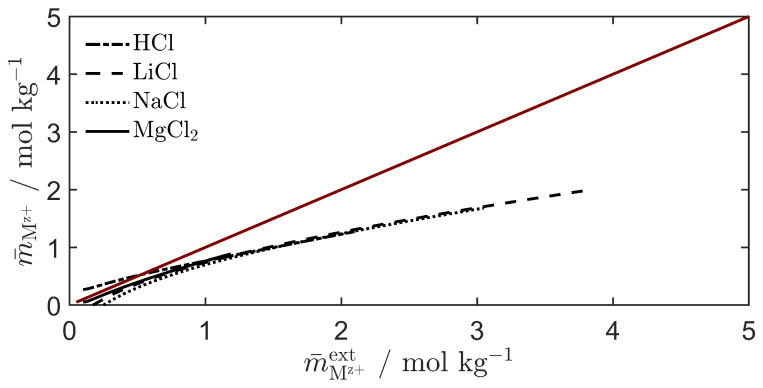
The estimated average cation molal concentration across the membrane, ${\bar{m}}_{M_{z+}}$, compared to the average external test solution concentration, ${\bar{m}}_{M_{z+}}^{\mathrm{ext}}$. Curves are generated using the apparent transport numbers of Figure 3 together with the ionic transport number and water transference coefficient from the linear region of curves from Figure 4. A solid red line shows the arithmetic mean of the external concentrations.

**Table 1 entropy-27-00075-t001:** The molal concentrations of the test solutions; the reference compartment molality is 0.1 mol ${\mathrm{kg}}^{-1}$ for all salts. Concentration cell measurements with KCl and the same cation-exchange membrane were taken from Solberg et al. [19]. Additional information for the relevant salts, such as thermodynamic data and calculated chemical potentials, is available in the Appendix A.

HCl	LiCl	NaCl	KCl [19]	MgCl_2_	CaCl_2_	NH_4_Cl
$m_{{\mathrm{MCl}}_{z+}}$ (${\mathrm{mol kg}}^{-1}$)						
0.2	0.5	0.5	0.5	0.2	0.2	0.2
0.5	1	1	1	0.5	0.5	0.5
0.9	2	2	2	1.0	1.0	1
1.7	4	4	3	1.9	2.0	2
4.1	6	6	4.7	4.8	5.8	6
-	8	-	-	-	-	-

**Table 2 entropy-27-00075-t002:** Ionic transport numbers, $t_{M^{z+}}$, and water transference coefficients, $t_{w}$, from the linear region of the permselectivity measured for the Selemion CMVN cation-exchange membrane. The reported values for LiCl, MgCl_2_ and CaCl_2_ are valid for test compartment salt concentrations below 3.5, 1.3 and 1.4 mol ${\mathrm{kg}}^{-1}$, respectively.

	$t_{M^{z+}}$	$t_{w}$
HCl	1.00 (±0.01)	1.1 (±0.8)
LiCl	1.00 (±0.01)	9.2 (±0.8)
NaCl	1.01 (±0.03)	4.9 (±0.2)
KCl	0.99 (±0.01)	3.7 (±0.4)
MgCl_2_	1.00 (±0.01)	8.5 (±0.5)
CaCl_2_	1.00 (±0.01)	6.2 (±0.6)
NH_4_Cl	1.00 (±0.01)	3.8 (±0.5)

## Data Availability

Dataset available on request from the authors.

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
