# Peer review of "Transport Numbers and Electroosmosis in Cation-Exchange Membranes with Aqueous Electrolyte Solutions of HCl, LiCl, NaCl, KCl, MgCl2, CaCl2 and NH4Cl"

_entropy, 2025, doi:10.3390/e27010075_

Round 1

Reviewer 1 Report

Comments and Suggestions for Authors

The article discusses the phenomenon of ion and water transport in systems with ion exchange membranes from the standpoint of chemical and non-equilibrium thermodynamics. The authors estimate the values of counterion and water transport numbers based on membrane potential measurements. For this purpose, the method of fitting experimental and calculated values is used. The work is relevant because it concerns transport phenomena in systems with ion-exchange membranes in chloride solutions of a wide range of cations for use in a promising method for obtaining electrical energy from a concentration gradient. The novelty is due to the use of a well-known approach to calculating counterion and water transport numbers for a wide range of electrolytes. The issue of the article's compliance with the subject matter of the journal is controversial.

There are questions and comments on the text of the article

1. The authors should more clearly indicate the novelty of the work in comparison with the regularities of the influence of the counterion nature on the permselectivity and electroosmotic permeability of cation-exchange membranes presented in the literature, for example [doi:10.1016/j.cis.2008.01.002, https://doi.org/10.1016/j.electacta.2003.11.030, https://doi.org/10.1016/j.memsci.2018.12.024, https://doi.org/10.1016/j.jcis.2004.04.066]. The number of self-citations should also be reduced.

2. When measuring the membrane potential, the authors study a 100 μm thick membrane surrounded by solutions whose concentrations differ in some cases by a factor of 10 or more. In this case, the diffusion flux through the membrane becomes significant. How is it taken into account? Did authors analyze its influence on concentration of dilute solution? To verify the calculation method, authors should compare the calculation results with the data of direct measurement of water transfer numbers or calculation of counterion transfer numbers based on other approaches.

3. Most often, the coefficient tw is called the water transfer number, while the authors use the water transference coefficient [Suhara M., Oda Y. Transport Number through the Perfluorinated Cation Membrane, Flemion //Proceedings of the Symposium on Ion Exchange--Transport and Interfacial Properties. – Electrochemical Society, 1981. – Т. 81. – №. 2. – С. 290. https://doi.org/10.1016/j.electacta.2003.11.030].

4. Formula 6 is a variant of the well-known Scachard equation, which is not mentioned in the article.

5. Page 3: line 81. The formula for the total flux should probably have “+”, otherwise the value calculated by the formula 2 is not the cation transfer number.

Author Response

Reviewer #1: The article discusses the phenomenon of ion and water transport in systems with ion exchange membranes from the standpoint of chemical and non-equilibrium thermodynamics. The authors estimate the values of counterion and water transport numbers based on membrane potential measurements. For this purpose, the method of fitting experimental and calculated values is used. The work is relevant because it concerns transport phenomena in systems with ion-exchange membranes in chloride solutions of a wide range of cations for use in a promising method for obtaining electrical energy from a concentration gradient. The novelty is due to the use of a well-known approach to calculating counterion and water transport numbers for a wide range of electrolytes. The issue of the article's compliance with the subject matter of the journal is controversial. There are questions and comments on the text of the article

Comment 1: The authors should more clearly indicate the novelty of the work in comparison with the regularities of the influence of the counterion nature on the permselectivity and electroosmotic permeability of cation-exchange membranes presented in the literature, for example [doi:10.1016/j.cis.2008.01.002, https://doi.org/10.1016/j.electacta.2003.11.030, https://doi.org/10.1016/j.memsci.2018.12.024, https://doi.org/10.1016/j.jcis.2004.04.066]. The number of self-citations should also be reduced.

Response:  Thank you for bringing these relevant works to our attention, and we have edited the manuscript according to the suggestion of the reviewer. We have also added a paragraph to the results and discussion section to compare our results with these relevant studies from the literature.

Comment 2: When measuring the membrane potential, the authors study a 100 μm thick membrane surrounded by solutions whose concentrations differ in some cases by a factor of 10 or more. In this case, the diffusion flux through the membrane becomes significant. How is it taken into account? Did authors analyze its influence on concentration of dilute solution? To verify the calculation method, authors should compare the calculation results with the data of direct measurement of water transfer numbers or calculation of counterion transfer numbers based on other approaches.

Response: This issue is mentioned in the experimental section (Section 4). The measured electric potential was generally stable after an initial (first 15 minutes) small increase (100-1000 μV), even with only 1 ion-exchange membrane. This was not the case for HCl, where we immediately observed a decreasing trend in the measured potential because of a reduction in the concentration difference due to diffusion. We therefore used 10 membranes for HCl, and we then obtained the same stability as for the other salts. Ion-exchange membranes typically have smaller salt diffusion coefficients than aqueous mixtures (https://doi.org/10.1016/j.memsci.2009.05.047), and we estimate for a concentration difference of 10 mol/L, a membrane area of 7 cm2, a solution volume of 0.2 L:

That is, a 1 % change in the reference solution concentration (0.1 mol/L) for the a more extreme concentration difference than we have used.

A future study may want to verify the water transference coefficients obtained from the emf with those from other approaches such as the streaming potential method or by measuring concentrations over a period of time under current producing conditions. Future studies should also investigate the concentration dependence of the water transference coefficients. This is now properly discussed in Section 5.2, and a sentence has been added to Section 6 on the topic of future studies.

Comment 3: Most often, the coefficient tw is called the water transfer number, while the authors use the water transference coefficient [Suhara M., Oda Y. Transport Number through the Perfluorinated Cation Membrane, Flemion //Proceedings of the Symposium on Ion Exchange--Transport and Interfacial Properties. – Electrochemical Society, 1981. – Т. 81. – №. 2. – С. 290. https://doi.org/10.1016/j.electacta.2003.11.030].

Response: It is true that the coefficient tw is often referred to as the water transfer number (or water transport number and water transference number). We wish to follow the nomenclature of Kjelstrup and Bedeaux (specifically the book Non-Equilibrium Thermodynamics of Heterogeneous Systems), in which tw is referred to as a transference coefficient and not a transport number. The reason is that we use transport numbers for charged species to describe the fraction of electric current carried by that species, while transference coefficients describe the amount of a neutral component carried along indirectly by the electric current. We find the distinction useful and have edited the manuscript to further explain the naming convention.  

Comment 4: Formula 6 is a variant of the well-known Scachard equation, which is not mentioned in the article.

Response: We are not familiar with the custom of referring to Equation 6 as the “Scatchard equation”. However, we have added a reference to this naming convention beneath the equation.

Comment 5: Page 3: line 81. The formula for the total flux should probably have “+”, otherwise the value calculated by the formula 2 is not the cation transfer number.

Response: Thank you for pointing this out. We did indeed find that a Faraday constant was missing from the current density definition on line 81.

Reviewer 2 Report

Comments and Suggestions for Authors

Ion and water transport numbers are important parameters for describing transport phenomena in ion-exchange membranes. The article provides interesting and well-researched results regarding the transport numbers of a commercial cation-exchange membrane in different electrolytes. It is well-written and clearly presented. I would recommend its publication with some minor revisions and clarifications:

-In a highly selective cation-exchange membrane, the transport of Cl ions can be considered negligible. Thus, Figure 1 might be misleading. The arrow representing the transport of Cl ions through the membrane could, for example, be shown with dotted lines to indicate that co-ions are repelled by the membrane.  

-The authors have been somewhat vague regarding Equation (8). Please, could  provide a more detailed discussion of this equation and include an appropriate reference?

-Although the information is presented in the supplementary material, the values of water chemical potential could be included in Table 1. Unlike the chemical potential of ions, these values are not shown in any table or figure in the text.

-The values corresponding to CaCl₂ and NH4Cl are not presented in Figures 2 and 3. Why? Results for these two electrolytes are instead provided in Table 2 and Figures 4 and 5.

-How are temperature and pressure maintained constant?

-In the last paragraph of Section 4, diffusive mixing is discussed. What about the osmotic effect? Some mention of this effect should be included in the text.

-More discussion should be provided to explain how the average cation concentration across the membrane has been estimated.

-What do the authors think is the cause of the low water transport number obtained for HCl?

Author Response

Reviewer #2: Ion and water transport numbers are important parameters for describing transport phenomena in ion-exchange membranes. The article provides interesting and well-researched results regarding the transport numbers of a commercial cation-exchange membrane in different electrolytes. It is well-written and clearly presented. I would recommend its publication with some minor revisions and clarifications:

Comment 1: In a highly selective cation-exchange membrane, the transport of Cl⁻ ions can be considered negligible. Thus, Figure 1 might be misleading. The arrow representing the transport of Cl⁻ ions through the membrane could, for example, be shown with dotted lines to indicate that co-ions are repelled by the membrane.  

Response: Thank you for the suggestion. We have updated the figure accordingly.

Comment 2: The authors have been somewhat vague regarding Equation (8). Please, could provide a more detailed discussion of this equation and include an appropriate reference?

Response: We thank the reviewer for the suggestion and have rewritten the section accompanying Equation 8 to better communicate the idea behind the choice of the fitting function. The function itself has not been taken from the literature.  

Comment 3: Although the information is presented in the supplementary material, the values of water chemical potential could be included in Table 1. Unlike the chemical potential of ions, these values are not shown in any table or figure in the text.

Response: Thank you for the suggestion. We think the current iteration of Table 1 provides adequate information for interpretation of the results, and that additional information on the chemical potentials would clutter the table.   

Comment 4: The values corresponding to CaCl₂ and NH4Cl are not presented in Figures 2 and 3. Why? Results for these two electrolytes are instead provided in Table 2 and Figures 4 and 5.

Response: The results for KCl, NH4Cl and CaCl2 were left out of Figures 2 and 3 to improve readability, as mentioned at the start of Section 5.1. We deem the results for these three salts to be similar enough to the results shown in Figures 2 and 3 that they do not need a separate figure.  

Comment 5: How are temperature and pressure maintained constant?

Response: We have revised the experimental section to clarify that the isobaric condition is maintained by controlling that the height of the two electrolyte solutions are at the same level, and that the experiments are performed at room temperature. The solutions were neither heated nor cooled.  

Comment 6: In the last paragraph of Section 4, diffusive mixing is discussed. What about the osmotic effect? Some mention of this effect should be included in the text.

Response: Osmosis is the diffusive mixing of water (due to a chemical potential difference for water), and it would also be seen in the experiment as a decrease in the voltage over time. We have edited the description to further clarify, and the sentence now reads:
“A negative drift was generally not present in the measured potential, indicating that diffusive transport of both water (osmosis) and/or salt did not significantly alter the electrolyte solution concentrations over the course of the measurement.

Comment 7: More discussion should be provided to explain how the average cation concentration across the membrane has been estimated.

Response: Thank you for pointing out this short-coming. We have added a paragraph to Section 3.2 to clarify how this mean concentration is estimated and what it represents.

Comment 8: What do the authors think is the cause of the low water transport number obtained for HCl?

Response: We thank the reviewer for the intriguing question. The results are very similar to those reported by Okada et al. (https://doi.org/10.1016/S0013-4686(98)00132-7) for the Nafion 117 membrane, and we have added the following discussion of the results to the manuscript:  
“The water transference coefficient of H+,  tw = 1.1 ± 0.8, suggests that migrating protons generally enter and leave the membrane as H3O+ ions. Furthermore, the low magnitude is similar to results from Okada et al. for the Nafion 117, and suggests that proton migration in the Selemion CMVN occurs in part via the Grotthuss mechanism as reported for the Nafion 117.”

Reviewer 3 Report

Comments and Suggestions for Authors

The authors of the article did a very interesting and promising study. They show new measurements of concentration cells with the Selemion CMVN cation-exchange membrane and single-salt solutions of HCl, LiCl, NaCl, MgCl2, CaCl2 and NH4Cl. However, the presentation of the results and their interpretation leave much to be desired. Following are detail comments/suggestions:

Please provide more details on the experimental design, including the number of repetitions, estimation of experimental errors, and control of experimental conditions.

2. For the Selemion CMVN cation-exchange membrane used in the experiments, it is recommended to describe in detail its pre-treatment and equilibration process.

3. Please explain in detail the selection and fitting process of the non-linear regression models and how these models better reflect the experimental data. And the non-linear behavior observed in the experiments, it is suggested to provide more physical or chemical explanations and discuss possible influencing factors.

4. Whether the regression lines and confidence intervals in Figure 2 and 4 are clearly represented and ensure that all data points in the figures are clearly marked? And it is suggested to to visually display the changes in membrane performance under different salt solutions.

5. Please expand the discussion section to more deeply compare the effects of different salt solutions on membrane performance and discuss the reasons behind these differences. The relationship between the membrane's water transport coefficient is not clear for readers.

6. Please provide more details on the experimental setup and operating conditions, including the equipment used, specifications of materials, and specific steps of the experiments.

7. What methods were employed to quantify the uncertainty in the measurements? Were there any systematic errors considered, such as potential drift over time or variations in membrane properties? How were these errors propagated through the calculations of transport numbers and water transference coefficients?

8. The authors should be more and more carefully to check the spelling and superscript.

9.This manuscript is more about the introduction of mechanism, and the part involving application is less. It is hoped that the author can supplement the relevant application part to further refine the views. Please discuss the universality of the research results under different types of ion-exchange membranes and operating conditions.

Author Response

Reviewer #3: The authors of the article did a very interesting and promising study. They show new measurements of concentration cells with the Selemion CMVN cation-exchange membrane and single-salt solutions of HCl, LiCl, NaCl, MgCl2, CaCl2 and NH4Cl. However, the presentation of the results and their interpretation leave much to be desired. Following are detail comments/suggestions:

Comment 1: Please provide more details on the experimental design, including the number of repetitions, estimation of experimental errors, and control of experimental conditions.

Response: Thank you for the suggestions. We have edited the Experimental Methods accordingly (Section 4). Experimental errors are evaluated according to standard procedures: the experimental variance in molal concentration, temperature and the measured voltages (concentration cell and bias measurement over 1800 and 600 s) are combined with the regression variance of the activity coefficient through the propagation of error. This yields the variance of chemical potentials and permselectivity.   

Comment 2: For the Selemion CMVN cation-exchange membrane used in the experiments, it is recommended to describe in detail its pre-treatment and equilibration process.

Response:  The pre-treatment and equilibration process has been described in the second paragraph of the Experimental Methods (Section 4).

Comment 3: Please explain in detail the selection and fitting process of the non-linear regression models and how these models better reflect the experimental data. And the non-linear behavior observed in the experiments, it is suggested to provide more physical or chemical explanations and discuss possible influencing factors.

Response:  We thank the reviewer for the suggestion and have rewritten the section accompanying Equation 8 to better communicate the idea behind the choice of the fitting function. The function itself and the choice of a non-linear over linear regression model are not crucial, but being monotonically increasing is seen as important here. We note that the hyperbolic tangent function or error function can also be good regression models. Still, the permselectivity analysis, with the linear regression model, is in focus in this work due to how it yields the transport number and water transference coefficient.

Comment 4: Whether the regression lines and confidence intervals in Figure 2 and 4 are clearly represented and ensure that all data points in the figures are clearly marked? And it is suggested to to visually display the changes in membrane performance under different salt solutions.

Response: We have updated Figure 2 and 4 to increase the size of the scatter boxes in order to enhance readability. The apparent transport number and the permselectivity directly show the membrane performance. A permselectivity of e.g. 0.5 means that only half of the energy available from the chemical potential gradient of the salt can be extracted as electrical energy in reverse electrodialysis.

Comment 5: Please expand the discussion section to more deeply compare the effects of different salt solutions on membrane performance and discuss the reasons behind these differences. The relationship between the membrane's water transport coefficient is not clear for readers.

Response:  Thank you for the suggestions. We have expanded the discussion section accordingly.

Comment 6: Please provide more details on the experimental setup and operating conditions, including the equipment used, specifications of materials, and specific steps of the experiments.

Response:  The use of concentration cells is well described in the literature and electrochemistry books. We argue that the Experimental Methods (Section 4) describes the specifics of the materials, equipment and procedure to such a degree that the experiment is reproducible.

Comment 7: What methods were employed to quantify the uncertainty in the measurements? Were there any systematic errors considered, such as potential drift over time or variations in membrane properties? How were these errors propagated through the calculations of transport numbers and water transference coefficients?

Response: Electric potential drifts over time and the handling of it is described both in the Experimental Methods (Section 4) and the first paragraph of Section 5. As mentioned in the response to Comment 1, the variance in the transient voltage measurement is included in the error propagation calculations. The double standard deviations for electric potentials and permselectivities are so small that they fall within the boxes in the scatter plots. The variance shown for the transport numbers and water transference coefficients are the 95 % confidence intervals for the regression coefficients in the permselectivity analysis.    

Comment 8: The authors should be more and more carefully to check the spelling and superscript.

Response:  Thank you for the concern, we have carefully checked the manuscript for errors.

Comment 9: This manuscript is more about the introduction of mechanism, and the part involving application is less. It is hoped that the author can supplement the relevant application part to further refine the views. Please discuss the universality of the research results under different types of ion-exchange membranes and operating conditions.

Response: This work does indeed focus on the analysis method of the electric potentials obtained. We have now added a new paragraph to the start of Section 5 to discuss how this work can contribute to the development of general models for transport phenomena in ion-exchange membranes.  

Round 2

Reviewer 1 Report

Comments and Suggestions for Authors

The authors have made changes to the text according to the recommendations. The article can be published in its current version.

Reviewer 2 Report

Comments and Suggestions for Authors

The manuscript has been adequately  modified. I think it is can be accepted  in present form.

Reviewer 3 Report

Comments and Suggestions for Authors

The authors have addressed all my comments, so I recommend the manuscript to be accepted for publication.

Comments on the Quality of English Language

It is OK.